# A Dusty Road for Astronauts

**DOI:** 10.3390/biomedicines11071921

**Published:** 2023-07-06

**Authors:** Silvana Miranda, Shannon Marchal, Lina Cumps, Jenne Dierckx, Marcus Krüger, Daniela Grimm, Sarah Baatout, Kevin Tabury, Bjorn Baselet

**Affiliations:** 1Radiobiology Unit, Belgian Nuclear Research Centre SCK CEN, 2400 Mol, Belgium; silvana.ferreira.da.silva.miranda@sckcen.be (S.M.); sarah.baatout@sckcen.be (S.B.); kevin.tabury@sckcen.be (K.T.); 2Department of Biotechnology, Faculty of Bioscience Engineering, Ghent University, 9000 Ghent, Belgium; 3Department of Microgravity and Translational Regenerative Medicine, Otto von Guericke University, 39106 Magdeburg, Germany; shannon.marchal@med.ovgu.de (S.M.); marcus.krueger@med.ovgu.de (M.K.); daniela.grimm@med.ovgu.de (D.G.); 4Department of Astronomy, Faculty of Science, Katholieke Universiteit Leuven, 3000 Leuven, Belgium; 5Research Group “Magdeburger Arbeitsgemeinschaft für Forschung unter Raumfahrt- und Schwerelosigkeitsbedingungen” (MARS), Otto von Guericke University, 39106 Magdeburg, Germany; 6Department of Biomedicine, Aarhus University, 8000 Aarhus, Denmark

**Keywords:** lunar dust, regolith, toxicity, inflammation, Apollo program

## Abstract

The lunar dust problem was first formulated in 1969 with NASA’s first successful mission to land a human being on the surface of the Moon. Subsequent Apollo missions failed to keep the dust at bay, so exposure to the dust was unavoidable. In 1972, Harrison Schmitt suffered a brief sneezing attack, red eyes, an itchy throat, and congested sinuses in response to lunar dust. Some additional Apollo astronauts also reported allergy-like symptoms after tracking dust into the lunar module. Immediately following the Apollo missions, research into the toxic effects of lunar dust on the respiratory system gained a lot of interest. Moreover, researchers believed other organ systems might be at risk, including the skin and cornea. Secondary effects could translocate to the cardiovascular system, the immune system, and the brain. With current intentions to return humans to the moon and establish a semi-permanent presence on or near the moon’s surface, integrated, end-to-end dust mitigation strategies are needed to enable sustainable lunar presence and architecture. The characteristics and formation of Martian dust are different from lunar dust, but advances in the research of lunar dust toxicity, mitigation, and protection strategies can prove strategic for future operations on Mars.

## 1. Introduction

Lunar exploration has been a fascinating area of interest since the beginning of recorded human history. The first successful manned lunar mission was accomplished by the United States (US) in 1969 with the Apollo 11 mission [1]. Since then, several countries have shown interest in lunar exploration. The primary objectives of lunar exploration are diverse, including using the moon as a testbed for technologies and capabilities needed for long-term human space exploration (e.g., Mars), setting up lunar bases as ‘fueling stations’ for much longer missions, and simply performing fundamental research under space conditions or with regard to the moon itself [2].

The idea to utilize the moon as a crucial stepping stone towards further human space exploration has currently gained increasing support, partially in light of the potential for economic gain from space resource utilization and exploitation [3]. Such efforts are endorsed and mirrored in US Space Policy through, e.g., the Artemis program [4]. The rest of the world is similarly interested in the economic potential of space and the Moon’s pivotal role in achieving it, as indicated by the already 10+ states signatory to the Artemis Accords [5], the European Union’s (EU) participation in the Lunar Gateway within the Artemis framework, the European Space Agency’s (ESA) Space Resources Strategy [6] and Strategy for Science at the Moon [7], and the Russian–Chinese intent to collaborate on an International Lunar Research Station (ILRS) [8], to mention a few. 

Bearing in mind these recent developments, it is clear that man’s return to the Moon is imminent. A key issue to be tackled in the lunar environment, with regard to a prolonged human presence on the moon, is lunar dust. At the latest, after the Apollo missions, it was clear that lunar dust could be a significant nuisance during space missions. After each outdoor mission, the astronauts’ suits had significant dust deposits in the outer material that could not be brushed off. Astronaut Harrison Schmitt reported that the inside of the Apollo 17 Lunar Module was temporarily “full of dust”, including the atmosphere that the astronauts breathed [9]. The dust that the astronauts brought back from the lunar surface covered surfaces as well as the astronauts themselves. In the past, some Apollo astronauts have had adverse reactions to lunar dust. In 1972, Harrison Schmitt suffered a brief sneezing attack, red eyes, an itchy throat, and congested sinuses in response to lunar dust. Some additional Apollo astronauts also reported allergy-like symptoms after tracking dust into their vehicles [10]. The issue of lunar dust and the potential effects it might have on biological systems are strongly linked to the peculiar properties and behaviors exhibited by the material, which are, in turn, driven by its physicochemical make-up. 

This comprehensive review will present a summary of the established physicochemical characteristics of lunar dust and its diverse spectrum of impacts on human health. Furthermore, the existing and upcoming interventions to mitigate the health adversities stemming from lunar dust exposure will be ascertained. 

In contrast with the Apollo program, the current objective is not to perform brief forays into this rather unknown and hostile environment but to establish a long-term base that would be, perhaps intermittently, inhabited by human personnel. The establishment of such semi-permanent lunar bases will require the successful resolution of key technological and psychological issues that have already been identified. Amongst others, these include power generation, radiation, and meteoroid shielding, microgravity effect mitigation, in situ resource utilization (ISRU), subsurface mission capability, environmental (incl. temperature) controls, and life-support systems [11]. 

## 2. Lunar Dust Physicochemical Properties

### 2.1. Creation and Maturation of Lunar Dust

Lunar dust has a certain life cycle, which can be divided into its creation and its maturation. Due to the moon’s lack of a dense atmosphere, meteoroid impacts cannot be avoided, and the continuous bombardment of objects great and small takes its toll on the surface. Upon landing, the meteoroid causes mass displacement and creates a crater. Over time, due to these impacts, mass displacement and churning of surface material cause the lunar surface to be covered in the regolith, which is defined as a combination of heterogeneous, loose rocks and dust laying atop the lunar bedrock. The heat and pressures generated during impact typically result in the regolith or bedrock partially melting and welting together with rocks of potentially different origins.

Throughout lunar history, meteoroid impact has been instrumental in influencing the lunar surface, thus generating new regolith material or altering the existing layers. In the older lunar highlands, this layer was found to be 10–15 m thick, whereas the younger Marias see regolith up to a depth of 4–5 m. Underneath this layer, fractured but not yet ground or pristine bedrock can be found. The mass flux of meteoroids reaching the lunar surface is approximately 10^6^ kg/year and consists of particles with a size range of 10 nm to 1 mm [12]. The bulk of the meteoroids is in the 30–150 µm size range and consists of particles with an individual mass of 10^−10^–10^−8^ kg, which impact the surface at velocities of ~10–72 km/s, thus causing a mass of 10^3^–10^6^ times the impact mass to be ejected, depending on the substrate. Since the ejecta particles are smaller than the impactor, this results in the very fine component of the lunar regolith being generated [13]. Furthermore, these secondary particles get launched at high speed and can contribute to the formation of dust clouds around the moon, or even escape the lunar environment entirely if their speed is >2.4 km/s [14].

The lunar regolith material, once created, is altered over time due to the effects of space weathering events while it remains on or near the surface. While exposed, (micro) meteoroid impacts can further churn and pulverize the regolith grains, and radiation can implant elements, produce tracks, or cause radiation damage. Furthermore, the large thermal cycles on the moon also have an effect on the surface material. Regolith that has barely seen the surface is considered immature, whereas grains that have been extensively exposed and altered by space weathering events are considered mature. Meteoroid impacts have the most profound effect on the regolith maturation process. The radiation reaching the lunar surface is a second driver of regolith maturation processes and consists mostly of protons, with heavier atoms being present in amounts similar to those seen in the distribution throughout the cosmos [15].

### 2.2. Grain Shape

Due to the dynamic nature of their formation and maturation processes, lunar dust grains are fairly variable in shape and morphology (Figure 1). The grain surfaces are continuously altered in both constructive ways, e.g., by taking up impact-melted material splashing onto them, and destructive ways, e.g., by sputtering due to solar-wind ions. Earlier investigations of lunar dust particles found that they could be described as elongated and very angular, having starkly irregular surfaces [1]. This research was performed mostly on larger particles (>20 µm), with less work done on grains down to the µm-size range [16]. An important finding was the frequent presence of impact-derived glass mounds on the particle surfaces (Figure 1F). 

More recent research on smaller grains [16], performed in tandem with the novel grain size analysis mentioned in the previous section [17], was done using scanning electron microscopy (SEM), following a similar sample preparation protocol as for the grain size analyses. The investigated lunar soil samples include the same Apollo 11 and 17 samples as discussed above, as well as immature Apollo 12 (12,001), mature Apollo 15 (15,041), and mature Apollo 17 (79,221) samples, thus providing a wide range of lunar soils. The samples were respectively dry-sieved to 43 µm (10,084 and 70,051) and wet-sieved to 10 µm (12,001, 15,041, and 79,221) prior to use. Overall, particle morphology was divided into 5 possible classes, observable with SEM (Figure 1).

Glassy beads consisting of material that melted due to micrometeoroid impact and was quenched in flight, thus forming round and elongated glassy beads (Figure 1A);Angular shards, being glassy in nature and having formed from the breakage or crushing of larger glassy fragments, exhibit sharp edges and an elongated shape (Figure 1B);Particles with a vesicular texture, mostly grains with holes or vesicles, were likely left by solar wind volatiles escaping the structure (Figure 1C);Aggregated particles, loosely attached or other, larger grains (Figure 1D);Blocky fragments, consisting of broken minerals and rocks, with irregular edges due to their formation through breakage (Figure 1E).

### 2.3. Composition 

The chemical composition of lunar dust varies but can be summarized as 50% silicon dioxide (SiO_2_), 15% aluminum oxide (Al_2_O_3_), 10% calcium oxide (CaO), 10% magnesium oxide (MgO), 5% titanium dioxide (TiO_2_), 5–15% iron (Fe) [16]. The iron found on the moon is different from that found on Earth (Fe^3+^) due to the lack of oxygen. It is therefore mostly present in a reduced state [18].

The agglutinic glass, which makes up a large fraction of the lunar regolith and especially the smaller lunar dust portion thereof, is heterogeneous in composition and reflects the variety in the mineral components of the regolith. Agglutinates contain many vesicles and are highly irregular in shape due to the escape of gaseous solar wind elements such as hydrogen (H) and helium (He) during micrometeoroid impact [1]. As these gases heat up and escape, they may interact with iron oxides (FeO) present in the lunar dust and reduce it to metallic iron (Fe^0^), forming water in the process. The pure iron subsequently reorganizes into nanometer-size droplets, either suspended in the glassy matrix or exposed on the particle’s surface [14]. Such nanophasic, metallic Fe droplets (np-Fe^0^) are, similar to the glass in which they are contained, more prominently present in the smaller particles [19,20]. The np-Fe^0^ was additionally found to contribute strongly to the reactivity of the lunar dust [21] and convey specific magnetic and microwave-interactive properties [22]. Overall, it is likely that the majority of the smaller lunar dust particles consist mostly of impact glass and contain a sizeable amount of np-Fe^0^ globules.

## 3. Human Health Effects 

Linked to the physicochemical properties of the dust are its possible negative effects on human health. As mentioned above, astronauts who have visited the moon have had adverse reactions to the lunar dust. The main symptoms involved the respiratory system and the eyes. However, other organ systems might be at risk and are in need of a comprehensive evaluation. 

### 3.1. Lessons Learned from Earth

Studies on the interaction of human cells and organ systems with lunar dust particles are limited. However, numerous works with terrestrial dusts and certain dust-like nanoparticles can be drawn on. 

Pneumoconiosis is a general term for a group of lung diseases caused by the lung’s reaction to inhaled mineral dust particles. The main cause of pneumoconiosis is workplace exposure and is rarely related to environmental exposures [23]. Looking into the etiology and pathophysiology of the aforementioned pulmonary diseases can provide valuable insight into the immunological reaction of the lungs to foreign particles [24]. The bulk weight of lunar dust is composed of SiO_2_ (~50% of the weight) [25]. Silicosis is an interstitial lung disease caused by chronic exposure to airborne inorganic silica particles. Occupations such as quarrying, since quartz is the most common form of crystalline silica on Earth, have been recognized to increase the risk of silicosis [23]. This mechanism is likely to be important for lunar dust because of its high concentration of silica. With the discovery of FeO and submicroscopic magnetite particles in Chang’E-5 lunar soil [26,27], symptoms of pulmonary siderosis may also not be excluded. 

Particulate matter (PM) is known to cause skin barrier dysfunction. A study showed that particulate matter with a diameter of 2.5 µm (PM2.5) induced an increase in tumor necrosis factor alpha (TNF-α), subsequently causing a filaggrin (FLG) deficiency in the skin. The authors proposed that a compromised skin barrier due to PM2.5 exposure may contribute to the development and exacerbation of allergic diseases such as atopic dermatitis [28]. 

Pulmonary oxidative stress resulting from inhaled PM may lead to systemic inflammation and, subsequently, increased cardiovascular risk. The American Heart Association (2004) released a scientific statement on “Air Pollution and Cardiovascular Disease” and concluded that exposure to PM air pollution contributes to cardiovascular morbidity and mortality. PM in air pollution has been shown to impair vascular function, increase blood pressure (BP), promote thrombosis and impair fibrinolysis, accelerate the development of atherosclerosis, increase the extent of myocardial ischemia, decrease heart rate variability (HRV), and increase susceptibility to myocardial infarction [29].

### 3.2. Formation of Reactive Oxygen Species

Lunar dust is a particulate matter with peculiar properties that convey a certain type of reactivity. This reactivity is of great relevance to the toxicity of dust to living organisms. Due to the constant bombardment by micrometeoroids and cosmic radiation, the surfaces of lunar dust particles are highly irregular and exhibit both mechanically sharp edges and features conveying chemical reactivity. These include dangling bonds with unsatisfied valences, mostly due to Si-O breakage in the prominent SiO_2_ fraction of the dust, and the implantation or exposure of reactive elements on the surface, such as np-Fe^0^. On Earth, the reactivity can be mitigated over time by the influence of oxygen and water in the atmosphere, but in the lunar vacuum, such mitigation does not occur [30].

In addition to their reactivity, lunar dust surface charges, hydrophilicity, and the potential of a particle to form H-bonds also influence its capacity to adsorb or react with phospholipid layers and proteins, thus causing cell membrane damage or rupture and protein denaturation [31,32].

Lastly, the capacity of particles to adsorb endogenous and exogenous substances can further exacerbate the hydrophilicity and free radical-related toxicity mechanisms mentioned above. In the event that a particle accumulates endogenous Fe, the production of free radicals would increase, as would the negative effects. When exogenous substances such as hazardous chemicals (e.g., nitric oxides) are adsorbed, the particle can facilitate their transport into the body and thus potentially allow otherwise inaccessible toxicity pathways to be initiated [33]. 

### 3.3. Organ-Specific Health Effects 

#### 3.3.1. Respiratory Adverse Effects 

The lungs form the body’s interface with the external air environment and facilitate the transfer of gases to and from the blood via the alveoli. PM can enter the lungs through the throat and trachea, making its way past the bronchi and bronchioles and potentially all the way to the alveoli. According to Park et al. [17], particles that are larger than 20 μm are usually caught and expelled through the nasal passage, while particles that are larger than 10 μm tend to be deposited in the upper respiratory system. In contrast, particles that are smaller than 10 μm are generally considered to be inhalable. These inhalable particles can be categorized as ultrafine (UF) particles (<0.1 μm), fine particles (0.1–2.5 μm), and coarse particles (2.5–10 μm). The depth to which particles can penetrate and their potential impact are determined by their size. Generally, UF and fine particles smaller than 0.5 μm are transported by diffusion, while larger fine particles and smaller coarse particles ranging from 0.5 to 8 μm are usually transported by sedimentation. Larger, coarser particles tend to have an immediate impact and are not transported. Coarse particles are cleared out within approximately 24 h by coughing because they get trapped in the mucus present in the trachea and bronchioles [17,34].

Fine particles, which make up 95% of lunar dust, have the ability to penetrate deeper into the respiratory system, where they come into contact with the epithelial cells of the alveolar ducts. The clearance of this group of particles is usually done by coughing due to the increased mucous secretion and macrophage recruitment to the area, which takes vastly longer, with timescales of days to months [17,34,35].

UF particles (<0.1 µm) will be able to travel to the alveoli and alveolar sacs (Figure 2). Here they activate the immune response and are trapped by macrophages, but they can also enter blood vessels and be transported to other organs [17,34,35,36]. The degree of toxicity is influenced by both the location where particles are deposited and the time it takes for them to be removed [19].

The reduced gravity on the moon causes charged lunar dust particles to be electrostatically repelled, levitate, and float in the air, increasing the risk of inhalation [37,38]. Since on earth particles between 8 μm and 0.5 μm in diameter deposit primarily by gravitational sedimentation, aerosol transport and deposition in partial gravity are likely to be altered. In terms of the risks associated with exposure to this dust, the amount and site of deposition are important because they determine the residence time [39]. For large particles (∼5 μm), impaction results in an increased relative deposition in the upper airways, where clearance mechanisms are effective, but for smaller particles (∼1 μm), the suggestion is that alveolar deposition will be increased, raising the possibility that these particles will be retained in the lung for a longer period of time (increased residence time), enhancing their toxic potential [40].

In cases where insufficient clearance of the particles occurs, macrophages still present on site will activate transcription factors that cause the release of ROS and reactive nitrogen species (RNS), cytokines, growth factors, lytic enzymes, and chemotactic factors [18,38,39], which will, in turn, attract other immune cells, such as lymphocytes, to the site. 

While in the alveolar space and contact with macrophages or other cells found therein, foreign particles can produce toxic effects by triggering inflammation and by producing and triggering the release of free radicals from macrophages. During chronic inflammation, the sustained radical presence brought on by the reactions at the particle surface, as well as the continuous release from macrophages, can overwhelm antioxidant defenses found in the alveolar lining layer and cause damage to cell membranes, deoxyribonucleic acid (DNA), and proteins.

As mentioned in Section 3.2, one of the mechanisms of toxicity for lunar dust particles is the formation of ROS, for example, through the activation and overexpression of the nicotinamide-adenine dinucleotide phosphate oxidase 4 (*NOX4*) and 2 (*NOX2*) genes [41]. This impaired activity of antioxidant enzymes aggravates the overall oxidative status. As mentioned by Sun et al., the balance between oxidizing and anti-oxidizing factors is broken by lunar dust [41]. As observed by recent experiments with lunar dust simulants, the viability of lung epithelial cells also decreased after exposure, along with the occurrence of DNA damage. The authors postulate that the latter is due to ROS and oxidative stress [42]. Together with an increase in collagen caused by these ROS, this might lead to pulmonary fibrosis [41]. Lam et al. showed that the broncho-alveolar lavage fluid obtained from rats treated with lunar dust simulants has an increased concentration of inflammation biomarkers, neutrophil, and lymphocyte counts, and total protein (an indicator of cell membrane permeability). This study also showed the distribution of the particles in the rats’ lung parenchyma and concentration-dependent alterations that could evolve into fibrosis [43].

#### 3.3.2. Immunological Adverse Effects 

The immune system is a complex network of cells, chemicals, and processes that work together to defend the body against foreign antigens, including microbes, viruses, cancer cells, and toxins. It provides protection to various areas of the body, such as the skin, respiratory passages, and intestinal tract, through both structural and chemical barriers. 

The immunologic cascade of events in the case of lunar dust inhalation begins with macrophages releasing ROS, RNS, cytokines, and other factors that will recruit other immune cells to the site [19].

Sun et al. tested the effects of lunar dust simulant CAS-1 on the immune system of rats [41]. Their results revealed an increase in lymphocytes and neutrophils in the broncho-alveolar fluid of the animals in the exposed group. These particular groups of immune cells have been shown to increase in the context of pulmonary fibrosis [44]. In contrast, the number of macrophages was significantly lower in lunar dust-exposed rats, and the authors postulate that this shift in the macrophage/neutrophil ratio could indicate a delayed response of the neutrophils that accumulate in the lung. 

Interleukin 6 (IL-6) and TNF-α are two pro-inflammatory cytokines that are increased in in vitro studies with lunar dust simulant (LDS), which can lead to chronic inflammation [45]. TNF-α is produced by the lungs epithelial cells in the presence of silica nanoparticles [46] or in response to exposure to smoke particles [47]. This will cause eosinophils and neutrophils to migrate to the inflammatory site and enhance the inflammatory response. TNF-α will also act as a stimulus to induce IL-6 production by the tissue’s macrophages and lymphocytes. This plethora of factors will cause a positive feedback loop with further production of TNF-α and the stimulation of inflammation [41]. Moreover, TNF-α can also lead to the development of fibrosis through the recruitment of fibroblasts and collagen production (Figure 3) [48]. The inducible nitric oxide synthase gene (iNOS), which is associated with inflammation in the pulmonary system, is also found to be overexpressed in response to stimulation by TNF-α [421 This was also significantly increased in rats exposed to LDS [49].

Studies conducted with crystalline silica showed that because of the negative charge of the particles, they can induce apoptosis in human alveolar macrophages by binding to the scavenger receptors class A I and II on the surface of these macrophages [50]. A study by Lam et al. suggests that this effect is dose-dependent and promotes a shift towards a pro-inflammatory state of the macrophages, which can lead to chronic lung inflammation and subsequent fibrosis [51,52]. Lunar dust also decreases the viability of macrophages, the surviving fraction showed altered morphology towards the pro-inflammatory active state. Moreover, phase contrast imaging showed phagocytized internalized particles of lunar dust in the macrophages [50,53]. Several studies have shown altered immune function in astronauts due to their exposure to the space environment. Reduction of T-cell activation and function, altered leukocyte distribution, and altered cytokine profiles are the most commonly observed alterations, which can lead to deficient immune responses to pathogens in space [54]. This should also be considered when studying the immune system’s response to lunar dust. 

#### 3.3.3. Cardiovascular Adverse Effects

At present, research on the health effects of exposure to lunar dust focuses mainly on tissues in direct contact with the dust particles, e.g., the respiratory system, the skin, and the cornea. However, other tissues may be secondarily affected through either the translocation of the dust from the site of initial contact or by adverse effects propagated to distal sites by reactions produced in the primary tissue [55].

From what is known regarding particulate inhalation on earth, dust particles smaller than 100 nm are able to enter the pulmonary capillaries from the alveolar surface and enter the cardiovascular system [35]. Nanoparticles were found in the lymph nodes, spleen, heart, liver, and even the bladder and brain [35,56,57,58]. The pathways underlying the effects on the cardiovascular system (CVS) are complex and remain to be elucidated. Extrapolating findings from dust outbreaks from deserts on Earth, Dominguez-Rodriguez et al. showed how exposure to high Saharan dust concentrations (PM_10_ > 50 µg/m^3^) is associated with increased in-hospital mortality in patients with heart failure [59]. Stafoggia et al. found an association between a 10µg/m^3^ increase in desert dust and an increase in mortality and hospitalizations in southern Europe [60]. Similarly, a 5% increase in total ICU admissions was observed during days of dust storms in northern America [61]. However, PM-induced oxidative stress and local and systemic inflammation repeatedly emerge as potential mechanisms in the development of cardiovascular disease (CVD) [29]. A study by Rowe suggested similar effects on the CVS in susceptible individuals exposed to lunar dust and air pollution [62].

Lunar dust experiments on rats demonstrated LDS-induced systemic inflammation as an important cause of autonomic dysfunction and myocardial fibrosis. Findings suggest increased parasympathetic activity as shown by an increase in high frequency and root mean square of successive differences (RMSSD) (two indicators of parasympathetic activity) and a decrease in low frequency. In addition, the study observed a total increase in HRV, systolic blood pressure with irregularities in heart rate (HR), electrocardiogram abnormalities such as ST-segment depression, and elevated levels of C-reactive protein, a biomarker for CVD. The systemic inflammation markers lactate dehydrogenase (LDH), TNF-α, and IL-6 were also elevated [55]. The authors proposed two mechanisms of action. The first being the inflammatory mechanism in which PM irritates the lungs, causing a local and then a systemic inflammatory response [63,64]; the like of which is believed to exacerbate cardiovascular disease [65]. The dysfunction of the autonomic nervous system is proposed as the second mechanism, which manifests as irregularities in HR and BP [66,67]. However, the study demonstrated an increase in inflammatory lesions and myocardial fibrotic lesions with an increase in LDS exposure dose. *NOX4* mRNA, transforming growth factor β1 (*TGF1B*), and collagen type 1α1 (*COL1A1)* mRNA significantly increased in the myocardial tissue of rats after LDS exposure [55]. This finding indicates that myocardial dysfunction after LDS exposure involves an inflammatory mechanism and cannot be solely explained by an imbalance of the sympathetic and parasympathetic nervous systems.

It has been proposed that lunar dust toxicity is largely due to its high iron content, consisting of np-Fe^0^ particles. Rowe postulated the ‘Apollo 15 Space Syndrome’ in which severe fingertip pain during space walks provided a warning of ischemia in the absence of Angina. Irwin’s three-time lunar excursion was more hazardous than Armstrong’s single excursion, as shown in an extraordinary stress test for hypertension and stress test cyanosis of nail beds. The authors proposed that the blue fingertips during a stress test on return could be trapped venous blood secondary to oxidative stress-induced endothelial injuries [68].

#### 3.3.4. Dermal Adverse Effects

Skin also comes into direct contact with lunar dust. The skin serves as the main protective barrier, shielding against harmful substances or pathogenic infections and protecting the body from water loss. Lunar dust is known to be highly abrasive, so clothing contaminated with lunar dust has the potential for skin abrasion and suit-induced injuries. The main source of dust exposure in previous crewed lunar missions was following an extravehicular activity (EVA), when dust was brought inside the lunar capsule and adhered to extravehicular suits [69,70]. Skin exposure to lunar dust may also be of concern if the interior of the EVA suit becomes contaminated with lunar dust, in which case dermal abrasion may take place at sites where the suit rubs against the skin. 

In the past, many skin-related issues were reported not only by astronauts onboard the International Space Station (ISS) but also by astronauts from the Mir and Apollo missions. The most common clinical issues were erythema, peeling and dryness of the skin, burning, pruritus, increased sensitivity of the skin, thinning, and delayed wound healing [71].

The pervasive nature of lunar dust particles could lead to dermal abrasion, irritation, or penetration into existing wounds [72]. Dermal abrasion in combination with skin damage from pressure points on anatomical prominences (fingertips, knuckles, elbows, and knees) could result in the breakdown of the stratum cornea, the outermost layer of the skin, which is important for the barrier function of the skin. A transdermal-impedance technique was employed to measure the abrasive effect of LDS on pig skin, a high-fidelity model for human skin. Findings demonstrated damage to the dry stratum cornea. Skin abrasion irritates the dermal/water–vapor barrier (dermis), leading to dermatitis and/or sensitization of the skin [70]. The outer-most layers of the skin are composed of keratinocytes, a highly regenerative cell type that undergoes rapid proliferation and terminal differentiation. Keratinocytes, in turn, build up the epidermal layers that are supported by the dermis, a complex tissue composed of mainly fibroblasts. Findings on human keratinocytes and fibroblasts exposed to LDS have shown cytotoxic effects and indicated fibroblasts’ greater susceptibility to deleterious effects. A decrease in cell number was observed upon exposure of fibroblasts to LDS, reflecting cell death, whereas keratinocyte numbers remained steady. Inspection of the cytoskeleton demonstrated more severe impacts of LDS on fibroblasts. The cells lost their elongated, spindle-like shape and rounded up before cell death was induced by the particulate stressors. Keratinocytes, in contrast, maintained their normal morphology. Lastly, in line with the structural changes observed in the actin system, the membranes of cells in dust-exposed fibroblast cultures became leaky within a few hours, whereas keratinocytes maintained their full integrity for up to two days [69]. Moreover, regeneration of keratinocyte cultures from scratch wounds exposed to LDS clearly demonstrated decreased cells’ abilities to perform proper wound healing. This conclusion was reached from the observation that the cellular ability to close a scratched area in the cultures was impaired [69]. In line with this research is the study by Monici et al., who exposed dermal fibroblasts to LDS and demonstrated a decreased ability of fibroblasts to adhere to a substrate and migrate in response to a wound [73]. This data was explained by a downregulation of genes encoding cadherins and catenins, suggesting the impairment of adhesion plaques. The authors conclude that in the case of wounds, ulcers, or burns, contamination with lunar dust could impair wound healing since it impairs fibroblast function, which is essential for efficient tissue repair. However, another study exposed a skin explant with a central 3 mm epidermal wound to lunar dust, martian dust, and earth dust and demonstrated no significant differences regarding wound closure, proliferation, apoptosis, or tissue degradation between the three groups. They did observe a significant increase in pro-inflammatory markers (IL-6; matrix metallopeptidase-9) and a decrease in matrix metallopeptidase-2 in all dust samples compared to no dust controls [74]. 

#### 3.3.5. Ocular and Neuronal Adverse Effects

The eyes are another barrier to the environment that may come into direct contact with regolith dust particles, either by direct deposition or by contact with contaminated objects touching the eye (e.g., fingers). Based on some reports from astronauts of eye irritation after exposure to large amounts of lunar dust, the National Aeronautics and Space Administration’s (NASA) Lunar Airborne Dust Toxicity Assessment Group (LADTAG) initiated the first toxicity studies with real lunar dust [29]. The studies by Meyers et al. reported minimal irritancy potential of lunar dust in vitro [75]. In vivo Draize tests in a rabbit model confirmed that finding. After one hour, mild conjunctival reactions (erythema and edema) were observed but subsided within 24 h. In addition, no corneal scratching or adverse signs or symptoms were noted in the cornea, iris, or conjunctiva within 72 h of observation. Therefore, the lunar dust was initially classified as minimally irritating for the eyes and mainly as an acute nuisance for ocular exposure [75]. These results are consistent with those for volcanic ash (which often serves as a simulant for lunar dust) or desert sand [76,77]. After the eruption of Mount St. Helens in 1980, ophthalmologists found that ash particles were well tolerated by exposed individuals and caused acute irritation but no long-term effects on the eyes [76]. A later 10-year study of the effects of volcanic ash on the eyes of school children found that eye symptoms were more common in highly exposed areas but were limited to minor transient effects (redness, itching, and foreign body reactions) that were easily treated with eye drops [78].

Nevertheless, moon dust should not be considered harmless. Krisanova et al. reported the unique effect of lunar dust on glutamate binding to nerve endings [79]. This may negatively affect extracellular glutamate homeostasis in the central nervous system and lead to changes in ambient glutamate levels, which are extremely important for proper synaptic transmission. In addition, there is evidence from animal studies that UF particles (median particle diameter ≤0.1 µm) can be transported to the brain via the olfactory bulb [56], across the blood-brain barrier from the circulation, or to a lesser extent via the trigeminal or facial nerves [80].

The interaction between lunar dust particles and various cells and tissues, as well as the mechanisms involved in clearing out and responding to this exposure, has been described. Figure 4 shows a summary of these adverse effects in different organ systems of the human body.

## 4. Dust Mitigation and Countermeasures 

The negative effects of lunar dust are not limited to the human body. Instruments and materials are also heavily affected by dust exposure, with the main concerns related to false readings, loss of traction, thermal control issues, and accelerated abrasion. Dust mitigation is comprised of a variety of strategies that work on different levels in order to reduce the negative effects of exposure to particles on human materials, and technology [81].

### 4.1. Technical Countermeasures 

Taking into account their primary goal, dust mitigation technologies can be included in four main groups:Dust Generation Avoidance. Encompasses designs and methods that reduce dust generation, such as dust barriers, fixing the surfaces where rovers and other vehicles will traverse on the lunar surface, and reducing the travel speed;Passive Mitigation. Special coatings and finishes for the surfaces of instruments will help minimize the adhesion of the lunar dust particles; seals and barriers that prevent the dust from entering the internal mechanisms of certain instruments;Active Mitigation. Technologies that will remove dust that has already entered facilities or instruments range from vibrational strategies to pressurized gas, brushes, or magnetic rollers;Dust Tolerant Design. Designs and materials that are resistant to the abrasion caused by dust. For example, for bearings, the materials proposed for space applications are zirconia and silicon nitride (ceramic bearings), stainless steel bearings, and superconducting magnetic bearings.

The Apollo missions laid the groundwork for assessing the issues that lunar dust can cause for future human operations on the lunar surface. In order to prevent damage to the spacesuits, which can be life-threatening to the astronauts, new spacesuits for future missions have been developed [82,83]. The pressurized system prevents the contact of the particles against the skin (in contrast with vacuum-based systems from the Apollo era) [67]. Furthermore, new materials are to be used in future spacesuits in order to minimize the adherence of dust particles to the surface of the suit. Other proposals are to coat the suit with dust-repellant materials or limit the use of the suits [84]. One system developed by Manyapu et al. is the spacesuit integrated carbon nanotube dust ejection/removal (SPIcDER) system [85]. This is a self-cleaning suit that repels dust particles by forming an electrical field around the suit’s surface.

### 4.2. Biological Countermeasures 

#### 4.2.1. Pharmacological Countermeasures 

As described earlier, the inhalation of lunar dust particles could result in the production of ROS by activation and overexpression of the *NOX4* and *NOX2* genes, which are released during oxidative stress [41,54,55,86]. One of the approaches being studied is the use of antioxidants to counteract the effects of oxidative stress caused by exposure to lunar dust. Antioxidants are compounds that can neutralize ROS and prevent cellular damage. However, a more targeted approach could be to block the source of ROS through the inhibition of NOX4 and NOX2, which could be promising for the prevention of pulmonary fibrosis [87]. NOX4 has also been shown to have a crucial role in cardiomyocyte homeostasis and redox mechanisms, possibly leading to myocardial damage. Inhibiting the *NOX4* gene could prove beneficial to prevent or treat lunar dust-related cardiac impairment as well as lung damage [55]. NASA is exploring the use of antioxidants as a potential countermeasure to the health hazards associated with lunar dust exposure since studies have shown that the administration of antioxidants can reduce the effects of oxidative stress caused by exposure to environmental factors such as pollution and radiation. Another promising approach is correcting the Mg deficiencies observed in astronauts. Recent studies hypothesized that severe Mg deficits cause transferrin (an iron-active protein that attenuates the ROS generation from inorganic iron) to work insufficiently against these hydroxyl radicals [42,68]. The observed increase in apoptosis in macrophages by the activation of the scavenger receptors could also be mitigated by treating the cells with polyinosinic acid, inhibiting the receptor’s activation by the lunar dust particles [50,52].

#### 4.2.2. Crew Protection Strategies

In line with the 4-branch strategy for dust mitigation, several standard operation protocols should be put in place for the crew in order to protect them against lunar dust. For example, eye washing stations should be installed in case of eye contamination; additional personal protective equipment should be worn by crew members that wear contact lenses since particles trapped between the lens and the eye surface could increase the risk of lesions [75]. 

It is now known that the probability of contamination with lunar dust particles is high, but the extent of the effects of prolonged exposure is still being investigated. Furthermore, when inhaled, particles are eliminated slowly from the body, and the toxic effect would depend on the exposure time, concentration, and time of clearance. Due to this and the uncertainties in relation to the full extent of the effects, exposure should be limited to 0.3 mg/m³ for a mission duration of one to six months, based on an exposure time of six hours each day, five days a week [88,89]. Nonetheless, these estimations are calculated based on the effects observed in animal model studies. For more informed decision-making, Space Agencies will need to invest in improved analog models for these investigations [43].

NASA has been researching ways to mitigate the health hazards associated with lunar dust exposure. They are developing new materials for spacesuits and equipment that can reduce the accumulation of dust and improve their functioning on the lunar surface. They are also developing methods to monitor the exposure of astronauts to lunar dust and improve the clearance of dust particles from the respiratory system [90].

### 4.3. Follow-Up Measures 

So far, the main focus has been on preventive measures, mitigation procedures, analyzing exposure effects, and setting appropriate exposure limits. Follow-up measures for astronauts on their return to Earth should also be discussed in the present topic. Medical examination requirements so far consist primarily of laboratory testing of blood and urine samples to diagnose conditions and diseases such as anemia, infection, and thrombocytopenia. Tests to determine the astronaut’s general health status are included (e.g., elucidating the body’s electrolyte balance and lipid status), as are other focus areas such as the musculoskeletal system, dermatology, ophthalmology, and the cardiovascular system [91]. As of today, no standardized pulmonary health screening is in place, highlighting the need for standardized cardiorespiratory testing and further follow-up assessments.

## 5. Open Questions for Future Human Space Exploration

### 5.1. Chronic Effects and Chronic Exposure Effects

To date, there is no data available on the human health risks associated with chronic exposure to lunar dust. With only 12 astronauts exposed to lunar dust, a limited number were available for follow-up studies looking into the chronic effects. In an attempt to answer these open questions, insights can be gained from earth-based research. Sun et al. [55] demonstrated an increase in inflammatory lesions and myocardial fibrotic lesions with an increase in LDS exposure dose. *NOX4* mRNA, *TGF1B* mRNA, and *COL1A1* mRNA significantly increased in the myocardial tissue of rats after LDS exposure. All are important factors in the formation of fibrosis, which is often associated with chronic phases of inflammatory diseases [92]. Another important finding demonstrated more severe stress test hypertension and cyanosis of the nail beds in an astronaut who participated in three EVAs compared to an Apollo astronaut who only participated in one [68]. 

An additional open question is the link between lunar dust and the development of cancer. Dust-induced DNA damage has been less well studied but is an aspect of possible long-term significance to human health. DNA damage can cause both short-term and long-term problems and, in addition, can affect both the nuclear and mitochondrial genomes. Mutations in nuclear DNA may lead to cell death or cancer [93]. In an animal study, rats exposed to particles isolated from air pollution developed nuclear DNA mutations in their sperm [94]. A human lung cell line treated with various particulate materials showed DNA strand breaks and activated caspase 9, an enzyme released from mitochondria in a process of cell death [95]. As indicated, lunar dust particles give rise to an excessive cellular oxidative stress response in human cells. Oxidative stress on the cellular level is widely perceived as a key factor in cancer development [96]. Furthermore, due to their particular shape and persistent nature, lunar dust particles are likely to induce lung cancer in similar ways as silica particles during silicosis on Earth [97]. Lastly, exposure to lunar dust may also pose a potential risk of developing cancer due to the presence of certain chemical components, such as chromium, which are known to be carcinogenic [98]. However, the exact mechanisms underlying the carcinogenicity of lunar dust and the threshold levels of exposure required to trigger carcinogenesis are still not fully understood, and further research is necessary to fully elucidate these potential risks.

### 5.2. Individual Susceptibility to Lunar Dust 

The Apollo program included nine three-crew missions to the Moon, of which only six included a lunar landing and subsequent human dust exposure. A total of 12 astronauts walked on the moon’s surface. Baseline characteristics of these astronauts include age (mean of 39 years old), gender (12 males, 0 females), number of EVAs performed per astronaut (1-3 EVAs), and time per EVA (mean of 5.25 h). To this day, no clinical data exists on the effects of lunar dust on female health. However, some insights can be obtained from PM-related health effects. Evidence showed inconsistencies in differences in PM-related health effects by sex. Results from dosimetric studies demonstrate sex-related differences in the localization of particles when deposited in the respiratory tract and in the deposition rate due to differences in body size, conductive airway size, and ventilatory parameters. In addition, one clinical study reported a significantly greater decrease in blood monocytes, basophils, and eosinophils in females than in males after controlled exposures to UF elemental carbon, suggesting potential sex-related differences in subclinical responses upon PM exposure [39].

The Apollo astronauts who performed EVA activities on the surface of the Moon were between the ages of 37 and 41 years old. From epidemiological studies on PM-induced health effects, findings showed increased respiratory effects from short-term PM exposure of all size fractions in children (<18 years) compared with adults. In addition, an increased risk of cardiovascular morbidity with short-term PM exposure in older adults (>65 years old) was observed [39]. The stringent requirements related to the astronaut selection process rule out applicants older than 50 at the time of applying. The likelihood of applicants younger than 18 years of age being accepted is also low because of the intense professional requirements that astronauts need to fulfill.

Controlled human exposure and toxicological studies have demonstrated a variety of health effects in response to PM exposure. Some of these studies indicated that populations with certain characteristics may be disproportionately affected. Findings suggest that the presence of null alleles or specific polymorphisms in genes that mediate the antioxidant response, regulate folate metabolism, or regulate levels of fibrinogen may increase susceptibility to PM-related health effects. Dosimetric studies demonstrated that COPD patients have increased dose rates of fine and UF particles and impaired mucociliary clearance relative to age-matched healthy subjects. These findings suggested that individuals with COPD are potentially at greater risk of PM-related health effects [99]. Again, the stringent astronaut selection process rules out all physical and psychological incompetence. Typical medical and psychological health standards are that applicants should be in general good health (pass the European Part-MED, class 2 medical examination for ESA applicants), free from any disease, and have visual acuity in both eyes of 100% (20/20 vision) either naturally or after correction with glasses or contact lenses [100,101].

### 5.3. Combined Effects with Other Space Environmental Factors

The environmental hazards to human health on the lunar surface are not just related to the crew’s exposure to lunar dust. Altered gravity and higher ionizing radiation exposures are major concerns for the overall wellbeing of future lunar explorers and mission success (Figure 5). Limited knowledge is available in regard to what the combined effects of these exposures could be.

As mentioned in the previous section, the reduced gravity of the Moon can exacerbate the lunar dust effects since normal clearance mechanisms for foreign particles entering the airways have been developed in normal gravity conditions. This can increase exposure times and lead to an accumulation of particles and a higher probability of adverse effects [19].

Additionally, the increased levels of radiation on the surface of the moon should be taken into account when trying to understand the effects of lunar dust exposure. On the one hand, radiation alone has a detrimental effect on several human body systems, for example, the immune system or the skin, triggers late cardiovascular effects, causes neurodegenerative issues, and increases cancer risk [102,103]. It might be that cumulative exposure to lunar dust could accelerate some of these outcomes. Additionally, one must take into account the cosmic radiation interaction with the dust particles themselves and the generation of secondary particles, whose contribution to the absorbed dose is harder to model and predict. Regarding the latter, in an effort to comprehensively investigate the radiation environment on the Moon, the Cosmic Ray Telescope for the Effects of Radiation (CRaTER), is included on the Lunar Reconnaissance Orbiter (LRO). This instrument aims to evaluate the impact of ionizing energy loss caused by penetrating solar energetic protons (SEP) and galactic cosmic rays (GCR) on silicon solid-state detectors and synthetic analogs of human tissue, known as tissue-equivalent plastic (TEP) [104].

## 6. Research Focus and Future Perspectives

### 6.1. Advanced In Vitro Models

A main disadvantage of in vitro models is that they fail to replicate the conditions of cells in an organism. For this reason, more and more scientific teams have focused on using methods of tissue engineering or modern cell biology to develop human tissue models that are suitable for the investigation of defined mechanisms and form a bridge between in vitro and in vivo research. A 3-dimensional (3D) cell culture is an artificially created environment in which biological cells are permitted to grow or interact with their surroundings in all three dimensions. 3D cell cultures have shown improvements in studies targeting morphology, cell number monitoring, proliferation, response to stimuli, differentiation, drug metabolism, and protein synthesis [105]. Another common tool used in research is the use of animal models. Mouse models are commonly used in research to test new drugs and treatment strategies, but they raise ethical concerns. 3D culturing techniques have allowed researchers to model tissues and organs in order to perform drug treatment tests on them and limit the use of animal models [106].

Another emerging development is the organ/tissue-on-chip model. Organ/tissue-on-chip systems contain engineered or natural miniature tissues grown inside microfluidic chips. The chips are designed to control cell microenvironments, maintain tissue-specific functions, and provide the potential to noninvasively study organ physiology, tissue development, and disease etymology [107,108].

The comprehensive review from Yang et al. [109] on the use of organ-on-chip models for investigating the effects of PM on different organ systems also shows the potential of applying these technologies to lunar dust studies, bridging the in vitro and in vivo gap without referring to animal models that often show different responses to humans, such as immune response differences or organ-specific outcomes. Moreover, the scale of this technology allows them to be used in real space conditions, which could allow the study of concomitant space stressors such as radiation, altered gravity, and lunar dust exposure in one setting [110,111].

### 6.2. Importance of Lunar Dust Simulants

It can be concluded from the current review that effective dust mitigation protocols are required to establish a sustainable presence on the moon. This, however, requires extensive testing and experimentation. Lunar samples brought to Earth as part of the Apollo program are available only on a very limited scale. Therefore, planetary surface simulations have been developed that reflect either the physical, mineralogical, or chemical properties of the lunar regolith [112]. These simulants are created from geologic material collected on Earth, some containing different types of glass to mimic the glass and volcanic glass components on the lunar surface that were formed by impacts. Some of these simulants are designed to mimic the geochemical properties of specific regions of the lunar surface, such as the lunar highlands simulant 1 (LHS-1) or the lunar mare simulant 1 (LMS-1) [113] Another example is the NASA/USGS-Lunar Highlands Stillwater anorthosite deposit (NU-LHT), which conveys similar shape, abrasiveness, and composition of lunar dust based on Apollo 16 samples and can be used for general experimental contexts [114]. However, comparative studies between real lunar dust and lunar dust simulants indicate important differences. The morphology of simulants might be different from real lunar dust by exhibiting smoother outlines, less complex surface textures such as glass mounds or vesicles, and lacking np-Fe^0^ [16]. For this reason, a careful selection of simulants for each test is necessary. In addition, some simulants are in limited supply or not in production at all anymore.

### 6.3. Lessons for Mars

With the new objectives to extend human space exploration missions to the red planet, exposure to Martian dust cannot be ruled out, assuming that some dust and soil will be brought inside the Martian habitat by returning astronauts, as was the case during the Apollo missions to the Moon. However, even less factual data exists for Martian soil and dust compared to lunar dust. The only source of data comes from the analysis of meteorites from Mars that have landed on Earth and the chemical and spectroscopic measurements of Mars soil taken by the Viking and Mars Pathfinder missions [115]. Important differences exist between lunar and Martian dust particles, and therefore, extrapolating data from lunar dust toxicity research should be done with caution. For instance, Mars rovers investigating the planet’s geology found that Martian dust likely contained perchlorates, a chemical known to harm the human thyroid, as well as minerals such as gypsum that can build up in the lungs. On Earth, similar particles cause serious health conditions in coal miners. Rovers also found traces of even more dangerous heavy metals, such as carcinogenic hexavalent chromium [116]. Moreover, mars and the Earth’s moon have very different surface and atmospheric environments. The moon has no atmosphere, whereas the Martian atmosphere is highly rarefied. The moon can get very cold, as low as −240 °C in the shadows, while Mars varies between −20 and −100 °C. Martian regolith results from the impacts of more massive meteorites and even asteroids and is known to be the result of daily erosion from water and wind [117]. However, one thing they have in common is that their surfaces are covered with a relatively homogeneous layer of dust, dominated by electrostatic charge. The electrostatic environment on the moon is created by the solar wind, cosmic rays, and solar radiation itself, while the environment on Mars is likely created by dust storm-induced grain collisions in the dust-laden atmosphere [118].

## 7. Conclusions

The challenges of the harsh space environment and lunar dust need to be addressed before humans can establish a sustainable presence on the moon. However, continuous research and development can aim to overcome these challenges and unravel the unanswered questions of our celestial neighbor.

The current review summarizes all the known health effects related to lunar dust exposure. A total of 12 astronauts from NASA’s Apollo missions were exposed to the foreign dust particles and reported only mild, short-lived symptoms. However, the spectrum of health effects associated with high-dose acute exposure or chronic low-dose exposure is not yet well understood. The amount of research on lunar dust toxicity is increasing, which likely reflects the rising urgency of resolving or mitigating the dust problem in time for the new wave of astronauts that will shortly head for the moon.

In conclusion, lunar exploration is an exciting area of interest that can provide valuable insights into the history of the solar system. The future of lunar exploration looks promising, with space agencies and private space companies investing in developing technologies to enable safe lunar missions.

## Figures and Tables

**Figure 1 biomedicines-11-01921-f001:**
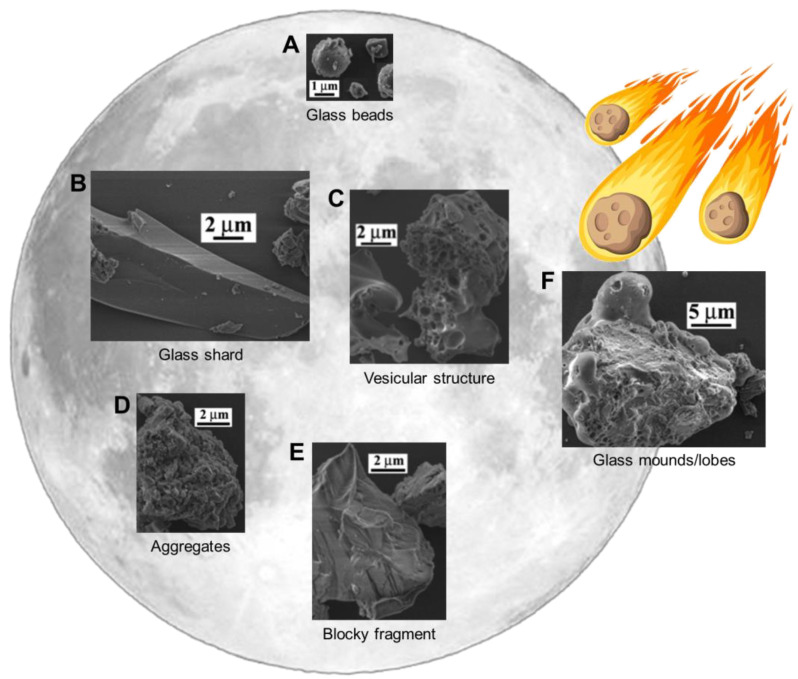
The 5 classes of Lunar dust grain shapes (**A**–**E**) and the glass coating that occurs due to impact-melt (**F**). Adapted from Liu et al., with permission [16].

**Figure 2 biomedicines-11-01921-f002:**
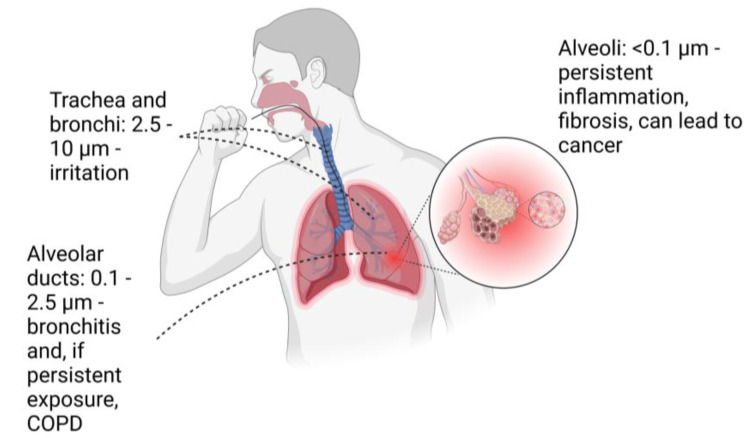
Particulate sizes and possible consequences depending on the location of the deposition. COPD-chronic obstructive pulmonary disease. Created with BioRender.com.

**Figure 3 biomedicines-11-01921-f003:**
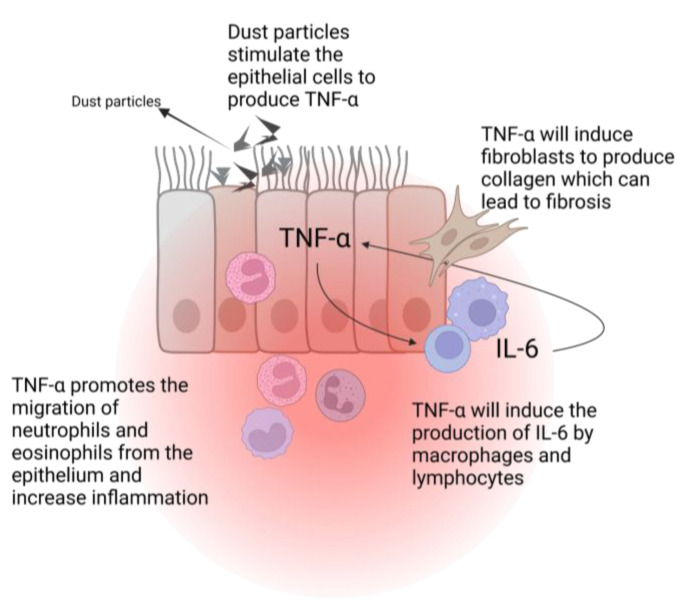
TNF-α induced pulmonary inflammation. IL-6: interleukin-6; TNF-α: tumor necrosis factor alpha. Created with BioRender.com.

**Figure 4 biomedicines-11-01921-f004:**
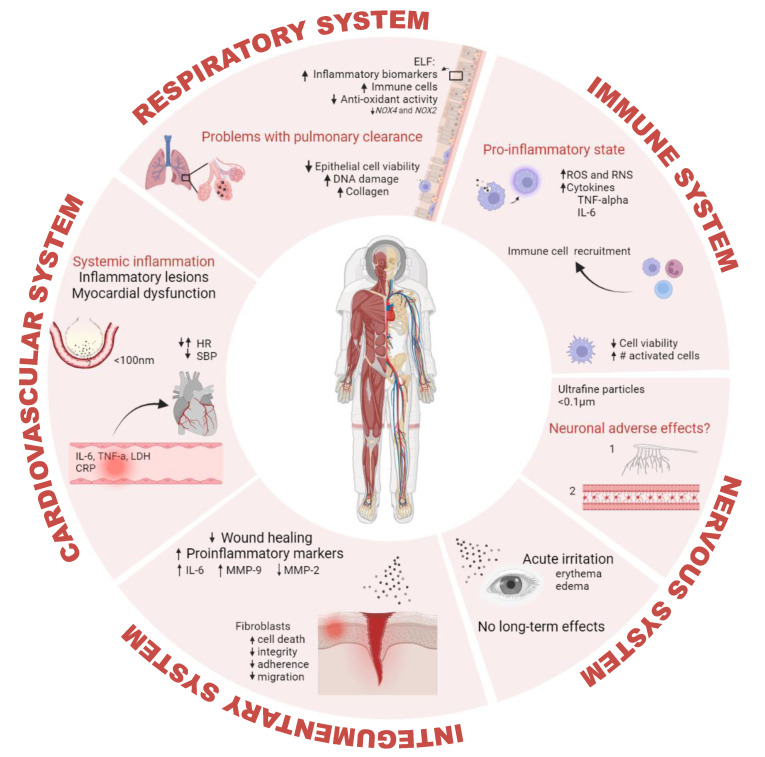
Lunar dust-induced human health effects. ELF: epithelial lining fluid; NOX4: Nicotinamide-adenine dinucleotide phosphate oxidase 4; NOX2: Nicotinamide-adenine dinucleotide phosphate oxidase 2; ROS: reactive oxygen species; RNS: reactive nitrogen species; TNF-alpha: tumor necrosis factor alpha; IL-6: interleukin 6; HR: heart rate; SBP: systolic blood pressure; LDH: lactate dehydrogenase; CRP: C-reactive protein; MMP-9: matrix metallopeptidase-9; MMP-2: matrix metallopeptidase-2; DNA: deoxyribonucleic acid. Created with BioRender.com.

**Figure 5 biomedicines-11-01921-f005:**
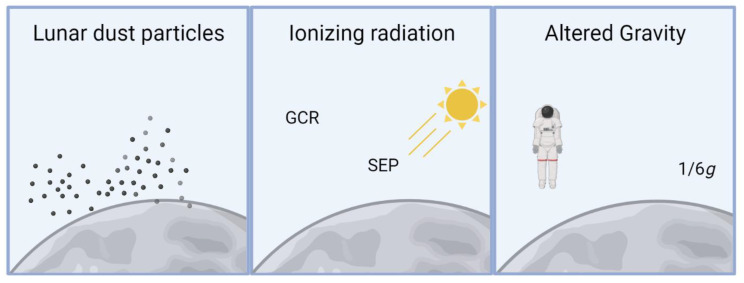
Additional space hazards on the moon impact human health. GCR: galactic cosmic rays; SEP: solar energetic particles. Created with BioRender.com.

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
