# Peer review of "A Dusty Road for Astronauts"

_biomedicines, 2023, doi:10.3390/biomedicines11071921_

Round 1

Reviewer 1 Report

Peer review of the document entitled: “A Dusty Road For Astronautsby Silvana Miranda and coworkers.

General comments:

The aim of the above-mentioned review was to summarize of the established physicochemical characteristics of lunar dust and its diverse spectrum of impacts on human health. The Authors describe the existing and upcoming interventions to mitigate the potential health adversities related to lunar dust exposure, concluding that the available research on lunar dust toxicity is progressively increasing, reflecting the urgency of ameliorating the dust problem for the new wave of astronauts that will shortly head for the Moon. The manuscript lays within the journal's scope.

Specific Comments:

Despite the work is properly organized, the main text could be shortened or suppressed, as Paragraph 6.3. Lessons for Mars (Lines 705-728).

In contrast some useful information regarding paragraph 3.3.3. Cardiovascular Adverse Effects (Line 341) may benefit from adding a few comments on former research on increased cardiovascular mortality rates during poor air quality events due to outbreaks of desert dust (please check the following references:

-Dominguez-Rodriguez A, et al. Saharan Dust Events in the Dust Belt -Canary Islands- and the Observed Association with in-Hospital Mortality of Patients with Heart Failure. J Clin Med. 2020 Jan 30;9(2):376. doi: 10.3390/jcm9020376. PMID: 32019177; PMCID: PMC7073718.

-Stafoggia M, et al. Desert Dust Outbreaks in Southern Europe: Contribution to Daily PM₁₀ Concentrations and Short-Term Associations with Mortality and Hospital Admissions. Environ Health Perspect. 2016 Apr;124(4):413-9. doi: 10.1289/ehp.1409164. Epub 2015 Jul 24. PMID: 26219103; PMCID: PMC4829979.

-Rublee CS,et al. Associations Between Dust Storms and Intensive Care Unit Admissions in the United States, 2000-2015. Geohealth. 2020 Aug 1;4(8):e2020GH000260. doi: 10.1029/2020GH000260. PMID: 32783014; PMCID: PMC7411550.

N/A

Author Response

General comments:

 The aim of the above-mentioned review was to summarize of the established physicochemical characteristics of lunar dust and its diverse spectrum of impacts on human health. The Authors describe the existing and upcoming interventions to mitigate the potential health adversities related to lunar dust exposure, concluding that the available research on lunar dust toxicity is progressively increasing, reflecting the urgency of ameliorating the dust problem for the new wave of astronauts that will shortly head for the Moon. The manuscript lays within the journal's scope.

Specific Comments:

  1. Despite the work is properly organized, the main text could be shortened or suppressed, as Paragraph 6.3. Lessons for Mars (Lines 705-728).

The authors thank the reviewer for the time and consideration in reading this manuscript. Overall the text was adapted in order to convey the information in a more succinct manner.

In the case of paragraph 6.3, the authors have added additional information to the abstract since it is maintained as a relevant section in the context of the lunar dust research efforts and the future of space exploration projects. Moreover, research shows similar effects in terms of inflammation response for both martian dust simulants and lunar dust simulants (1). This findings might highlight the utility of development of countermeasures for future Mars missions, that are not only technical but also biological, with the help of lunar based studies.

  1. Ahmadli G, Schnabel R, Jokuszies A, Vogt PM, Zier U, Mirastschijski U. [Impact of Martian and Lunar dust simulants on cellular inflammation in human skin wounds ex vivo]. Handchir Mikrochir Plast Chir. 2014;46(6):361-8. doi: 10.1055/s-0034-1394419

  1. In contrast some useful information regarding paragraph 3.3.3. Cardiovascular Adverse Effects (Line 341) may benefit from adding a few comments on former research on increased cardiovascular mortality rates during poor air quality events due to outbreaks of desert dust (please check the following references:

-Dominguez-Rodriguez A, et al. Saharan Dust Events in the Dust Belt -Canary Islands- and the Observed Association with in-Hospital Mortality of Patients with Heart Failure. J Clin Med. 2020 Jan 30;9(2):376. doi: 10.3390/jcm9020376. PMID: 32019177; PMCID: PMC7073718.

-Stafoggia M, et al. Desert Dust Outbreaks in Southern Europe: Contribution to Daily PM₁₀ Concentrations and Short-Term Associations with Mortality and Hospital Admissions. Environ Health Perspect. 2016 Apr;124(4):413-9. doi: 10.1289/ehp.1409164. Epub 2015 Jul 24. PMID: 26219103; PMCID: PMC4829979.

-Rublee CS,et al. Associations Between Dust Storms and Intensive Care Unit Admissions in the United States, 2000-2015. Geohealth. 2020 Aug 1;4(8):e2020GH000260. doi: 10.1029/2020GH000260. PMID: 32783014; PMCID: PMC7411550.

The authors acknowledge and appreciate the reviewer’s comment. The following was added to section 3.3.3 taking into consideration the references suggested by the reviewer. Thank you for this comment.

Lines 384-391:

“Extrapolating findings from dust outbreaks from deserts on earth, Dominguez-Rodriguez et al. showed how exposure to high Saharan dust concentrations (PM10>50µg/m3) is associated with increased in-hospital mortality in patients with heart failure [60]. Stafoggia et al. found an association between a 10µg/m3 in-crease in desert dust and an increase in mortality and hospitalizations in Southern Europe [61]. Similarly, a 5% increase in total ICU admissions was observed on days of dust storms in Northern America [62].”

 Comments can also be found in attachment

Reviewer 2 Report

The authors present a comprehensive analysis of the physical properties, health hazards, and mitigation strategies of moon dust.  This is overall a timely and comprehensive review with sufficient detail to capture the interest of diverse specialists.  The information content is overall excellent, but the presentation is awkward in places.  Most proffered comments are largely directed to developing a more rigorous presentation.  Major comments:

1.     The generally colloquial writing style is a bit jarring given the serious nature of the material and very substantial technical detail covering diverse scientific disciplines.  

a.     Suggest not writing in the first person, e.g., line 537, “We know that the probability…”

b.     Eliminate other colloquialisms, e.g., line 29, “since the dawn of time…”; line 202, cosmic radiation can substitute for “all kinds of ionizing radiation”; line 208, such mitigation does not happen can substitute for “this is evidently not the case”…; “on the other hand”; many other examples.

c.     Reduce use of the passive voice, i.e., line 385 can be replaced with “Skin also comes into direct contact with lunar dust”

2.     Please be consistent with citations.  In places, the format is Lam et al. [##] and others it is Lam et al….sentence….[##].  Either can be correct, please just use one format.

3.     Line 502-please elaborate on exactly what these interesting materials are.

4.     Table 1 contains a huge amount of largely useless information (only the 5 people well versed in synthetic moon dust will appreciate this information) that is only referred to once. Can probably save some ink and eliminate it, instead referring to a few of the references within the table as examples; comprehensiveness is not needed here.

5.     Section 6.3-please consider deleting this section on Mars dust which is out of place since the thesis was developed exclusively around moon dust.  If you insist on keeping this section, you need to refer to it in the abstract.

6.     Similarly, lines 210-217 can be safely eliminated because the oxidoreductive state of Earthly minerals is irrelevant to regolith.  

As above

Author Response

Reviewer 2 comments:

The authors present a comprehensive analysis of the physical properties, health hazards, and mitigation strategies of moon dust.  This is overall a timely and comprehensive review with sufficient detail to capture the interest of diverse specialists.  The information content is overall excellent, but the presentation is awkward in places.  Most proffered comments are largely directed to developing a more rigorous presentation. 

Major comments:

  1. The generally colloquial writing style is a bit jarring given the serious nature of the material and very substantial technical detail covering diverse scientific disciplines.
  2. Suggest not writing in the first person, e.g., line 537, “We know that the probability…”

The authors acknowledge and appreciate the reviewer’s comment. The authors have corrected the use of first person as suggested by the reviewer. Changes are marked with track changes in the manuscript. An example specific to this comment is line 569 (former line 537).

  1. Eliminate other colloquialisms, e.g., line 29, “since the dawn of time…”; line 202, cosmic radiation can substitute for “all kinds of ionizing radiation”; line 208, such mitigation does not happen can substitute for “this is evidently not the case”…; “on the other hand”; many other examples.

The authors acknowledge and appreciate the reviewer’s comment. Changes were made in all the text regarding the use of colloquialisms. These can be tracked  the use of track changes in MS Word. Specifically  first person as suggested by the reviewer. Changes are marked with track changes in the manuscript. Examples specific to this comment are:

  • Line 32 (former line 29): “Lunar exploration has been a fascinating area of interest since the course of recorded human history.”
  • Line 208 (former line 202): “Due to the constant bombardment by micrometeoroids and cosmic radiation, the sur-faces of lunar dust particles are highly irregular”
  • Line 214 (former line 208): “On Earth, the reactivity can be mitigated over time by the influence of oxygen and water in the atmosphere, but in the lunar vacuum such mitigation does not occur”

  1. Reduce use of the passive voice, i.e., line 385 can be replaced with “Skin also comes into direct contact with lunar dust”

The authors acknowledge and appreciate the reviewer’s comment. The use of passive voice was changed throughout the text (see track changes). The example referred to by the reviewer can be found in line 403, former line 385: “Skin also comes into direct contact with lunar dust.”

  1. Please be consistent with citations. In places, the format is Lam et al. [##] and others it is Lam et al….sentence….[##].  Either can be correct, please just use one format.

The authors acknowledge and appreciate the reviewer’s comment. Changes were made throughout the text in order to have an uniform manner of citing.

  1. Line 502-please elaborate on exactly what these interesting materials are.

Following the reviewer’s comments, the authors have added the additional information that can be found hereunder. Many thanks for this comment.

Line 523 (former line 502): “4.   Dust Tolerant Design. Designs and materials that are resistant to the abrasion caused by dust. For example for bearings, the materials proposed for space applications are zirconia and silicon nitride (ceramic bearings), stainless steel bearings, and superconducting magnetic bearings.”

  1. Table 1 contains a huge amount of largely useless information (only the 5 people well versed in synthetic moon dust will appreciate this information) that is only referred to once. Can probably save some ink and eliminate it, instead referring to a few of the references within the table as examples; comprehensiveness is not needed here.

Many thanks for this comment. The authors agree that the table 1 with the extensive types of lunar dust simulants does not have added value in the context of this manuscript. Therefore, the table has been removed and the text was adapted accordingly:

Section 6.1, lines 723-741:

“We can conclude from the current review that effective dust mitigation protocols are required to establish a sustainable presence on the Moon. This, however, requires extensive testing and experimentation. Lunar samples brought to Earth as part of the Apollo program are available only on a very limited scale. Therefore, planetary surface simulations have been developed that reflect either the physical, mineralogical, or chemical properties of lunar regolith (Table 1) [112]. These simulants are created from geologic material collected on Earth, some containing different types of glass to mimic the glass and volcanic glass components on the lunar surface that were formed by impacts. Some of this simulants are design to mimic the geochemical properties of specific regions of the lunar surface, such has the Lunar Highlands simulant 1 (LHS-1) or the lunar mare simulant 1 (LMS-1) [113]. Another example is the NASA/USGS-Lunar Highlands Stillwater anorthosite deposit (NU-LHT), which conveys similar shape, abrasiveness and composition of lunar dust, based on Apollo 16 samples and can be used for general experimental contexts [114]. However, comparative studies between real lunar dust and lunar dust simulants indicate important differences. The morphology of simulants might be different from real lunar dust by exhibiting smoother outlines, less complex surface textures such as glass mounds or vesicles, and lacking np-Fe0 [16]. For this reason, a careful selection of simulants for each test is necessary. In addition, some simulants are in limited supply or not in production at all anymore.”

  1. Section 6.3-please consider deleting this section on Mars dust which is out of place since the thesis was developed exclusively around moon dust. If you insist on keeping this section, you need to refer to it in the abstract.

The authors thank the reviewer for this comment. This information has been added to the abstract as the authors sustain that it is a relevant section in the context of the lunar dust research efforts and future of space exploration projects.

The amended abstract can be found hereunder:

“The lunar dust problem was first formulated in 1969 with NASA’s first successful mission to land a human being on the surface of the Moon. Subsequent Apollo missions failed to keep the dust at bay, thus exposure to the dust was unavoidable. In 1972, Harrison Schmitt suffered a brief sneezing attack, red eyes, an itchy throat and congested sinuses in response to lunar dust. Some additional Apollo astronauts also reported allergy-like symptoms after tracking dust into the lunar module. Immediately following the Apollo missions, research into the toxic effects of lunar dust on the respiratory system gained a lot of interest. Moreover, researchers believed other organ systems might be at risk, including the skin and cornea. Secondary effects could translocate to the cardio-vascular system, the immune system and the brain. With current intentions to return humans to the Moon and establish a semi-permanent presence on or near the Moon’s surface, integrated, end-to-end dust mitigation strategies are needed to enable sustainable lunar presence and architecture. The characteristics and formation of martian dust are different from lunar dust but advances in the research of lunar dust toxicity, mitigation and protection strategies can prove strategic for future operations on Mars.”

  1. Similarly, lines 210-217 can be safely eliminated because the oxidoreductive state of Earthly minerals is irrelevant to regolith.

Many thanks for this comment. The authors agree that the sentence paragraph regarding the Earth’s minerals oxireductive state can be omitted from the text. Changes were made accordingly and can be tracked with the track changes function on MS Word.
